# Integration of simulated annealing into pigeon inspired optimizer algorithm for feature selection in network intrusion detection systems

Wanwei Huang[1], Haobin Tian[1], Sunan Wang[2], Chaoqin Zhang[3] and Xiaohui Zhang[4]

[1] College of Software Engineering, Zhengzhou University of Light Industry, Zhengzhou, Henan, China
[2] Electronic & Communication Engineering, Shenzhen Polytechnic School, Shenzhen, Guangdong, China
[3] College of Computer and Communication Engineering, Zhengzhou University of Light Industry, Zhengzhou, Henan, China
[4] Henan Xinda Wangyu Technology Co. Ltd, Zhengzhou, Henan, China



Corresponding author
Sunan Wang,
wangsunansps@163.com

## ABSTRACT

In the context of the 5G network, the proliferation of access devices results in heightened network traffic and shifts in traffic patterns, and network intrusion detection faces greater challenges. A feature selection algorithm is proposed for network intrusion detection systems that uses an improved binary pigeon-inspired optimizer (SABPIO) algorithm to tackle the challenges posed by the high dimensionality and complexity of network traffic, resulting in complex models, reduced accuracy, and longer detection times. First, the raw dataset is pre-processed by uniquely one-hot encoded and standardized. Next, feature selection is performed using SABPIO, which employs simulated annealing and the population decay factor to identify the most relevant subset of features for subsequent review and evaluation. Finally, the selected subset of features is fed into decision trees and random forest classifiers to evaluate the effectiveness of SABPIO. The proposed algorithm has been validated through experimentation on three publicly available datasets: UNSW-NB15, NLS-KDD, and CIC-IDS-2017. The experimental findings demonstrate that SABPIO identifies the most indicative subset of features through rational computation. This method significantly abbreviates the system's training duration, enhances detection rates, and compared to the use of all features, minimally reduces the training and testing times by factors of 3.2 and 0.3, respectively. Furthermore, it enhances the F1-score of the feature subset selected by CPIO and Boost algorithms when compared to CPIO and XGBoost, resulting in improvements ranging from 1.21% to 2.19%, and 1.79% to 4.52%.

## INTRODUCTION

As 5G networks continue to advance and the number of access devices increases, network traffic has also increased significantly. With higher bandwidth, lower latency, and greater connection density, 5G networks are more vulnerable to insidious and efficient network attacks. To address network security concerns, it is recommended to implement a network intrusion detection system (NIDS) (*Tsai & Lin, 2010*) on computer systems to scan for any signs of unauthorized intrusion. The connection of a large number of devices to the 5G network requires NIDS to be capable of handling such a large-scale operation. Nevertheless, network data is characterized not only by its substantial volume but also by its high dimensional nature (*Ganapathy et al., 2013*), resulting in prolonged model training times and diminished predictive performance (*Hastie et al., 2009*). Hence, the significance of feature selection algorithms in NIDS is self-evident (*Alazab et al., 2012*). Feature selection offers a means of identifying significant features and eliminating extraneous ones from a dataset. The objective is to choose the most indicative subset of features from the initial dataset, thereby reducing model complexity and enhancing predictive performance. Feature selection reduces model complexity, improves predictive performance, and enhances the accuracy and reliability of intrusion detection by minimizing false alarms and preventing missed alarms (*Thakkar & Lohiya, 2022*). NIDS that uses feature selection algorithms have been extensively researched and implemented. They are a critical technical tool for ensuring network security.

The aim of feature selection is to identify a subset of features that closely approximates the optimal feature subset within a reasonable timeframe. The inclusion of feature selection has greatly improved the effectiveness of NIDS, aiming to identify a more suitable solution rather than an optimal one. At present, bio-inspired algorithms utilizing feature selection techniques exhibit superior performance when compared to other methods. Bionic algorithms draw inspiration from the collective behaviors of various animals (such as fireflies, wolves, fish, and birds), and researchers have introduced diverse computational approaches to emulate these species' behaviors for problem optimization, known as foraging. These approaches include the Chaotic Firefly Algorithm, Grey Wolf Optimization, Artificial Fish Swarm Algorithm, and the Bird Swarm Algorithm, among others (*Shoghian & Kouzehgar, 2012*). Each member within a swarm intelligence algorithm embodies a potential solution, generating fresh individuals through continuous mutation and crossover. The Pigeon-Inspired Optimizer (PIO) algorithm is an emerging swarm intelligence algorithm, which has obvious advantages in global search ability, convergence speed and robustness compared with other swarm intelligence algorithms.

Effective feature selection algorithms can enhance the detection capabilities and efficiency of NIDS. Scientific and efficient decision-making in feature selection has emerged as a critical method to guarantee the operational security of networks. However, feature selection algorithms currently face several issues, including excessive feature pruning (*Zhou et al., 2020*), disregard for inter-feature correlations (*Li et al., 2020*), susceptibility to anomalous traffic, and difficulty in handling large datasets (*Jaw & Wang, 2021*). These challenges can lead to a decline in the model's generalization ability, increased

complexity, and reduced stability, ultimately impacting the model's detection performance and efficiency (*Rashid et al., 2022*). To tackle the aforementioned issues, this article presents a feature selection algorithm for NIDS based on an improved binary pigeon-inspired optimization algorithm, aiming to enhance the accuracy and efficiency of feature selection in the context of network intrusion detection. The goal is to reduce false positive rate and false negative rate in NIDS. This approach utilizes mutation and simulated annealing mechanisms during the map and compass operator phases to expand the search scope and prevent the feature subset from being stuck in local optima. Furthermore, it introduces a population decay factor in the landmark operator phase to control rapid population decline and regulate the algorithm's convergence rate. The article presents a method that selects the most representative feature subset through reasonable computation. This leads to a significant reduction in model training and testing times, while enhancing the model's detection rate and accuracy. The key contributions of this study include:

(1) We conduct an investigation and analysis of existing NIDS feature selection algorithms, leading to the proposal of an improved NIDS feature selection algorithm based on enhancements to the binary PIO algorithm;

(2) during the map and compass operator phase, a mutation mechanism is introduced to increase the diversity of the population, thereby expanding the search space of the algorithm. Additionally, a simulated annealing approach is incorporated to accept new solutions that are worse than the current solution with a certain probability, facilitating escape from local optima;

(3) during the landmark operator phase, a population decay factor is proposed to dynamically adjust the population size for each iteration based on the fitness distribution of the population. The objective of this adjustment is to regulate the convergence speed of the algorithm;

(4) the improved PIO algorithm was combined with a classifier and applied to NIDS. The algorithm was evaluated against state-of-the-art feature selection algorithms using datasets such as UNSW-NB15, NSL-KDD, and CIC-IDS-2017.

The remaining sections of this article are organized as follows. "Related Work" provides an overview of previous related work conducted by other researchers. In "Continuous Pigeon Inspired Optimizer", we present the architecture and formulation description of the continuous PIO algorithm. "Proposed Improvement of PIO" describes the model of the proposed feature selection algorithm and provides detailed information on the updating steps. In "Experiments and Results", we conduct simulation experiments and evaluate the performance of our approach. Finally, in "Conclusion", we conclude and discuss future research directions.

## RELATED WORK

The classification performance of network intrusion detection system models is significantly constrained by the high dimensionality and sheer volume of network traffic data. In light of the increasing volume of data, researchers have investigated sample selection methods to enhance the efficiency of the training model process. Feature

selection algorithms have been developed to tackle challenges associated with high data dimensionality, as well as the presence of irrelevant and redundant features (*Alazab et al., 2012*) in datasets. Feature selection is crucial for enhancing model performance by eliminating irrelevant and redundant information from the dataset. By selecting only the most significant features for model training, it helps prevent overfitting and reduces feature dimensionality, thereby improving the efficiency of model training and prediction processes.

Traditional feature selection algorithms can be categorized into three types: filtered, wrapper, and embedded methods (*Di Mauro et al., 2021*). Filtered feature selection operates independently of the classifier, while wrapper methods involve evaluating the classifier during feature selection. Embedded methods integrate feature selection directly into the training process of the classifier. Each type has distinct benefits and is appropriate for different situations depending on the specific needs of the task. Filtered algorithms are computationally efficient but do not guarantee optimal feature selection. Embedded algorithms perform feature selection during intrusion model training and are computationally expensive for large datasets. Conversely, wrapper algorithms exhibit higher accuracy than the previous two algorithms but are sensitive to the quality of the training data. Achieving high accuracy is crucial for NIDS, and training time for offline data is not a significant concern. Therefore, this article uses the wrapper algorithm as the preferred method for feature selection, as it has been shown to provide the best results.

Table 1 provides a summary of the performance of different feature selection methods on different datasets, categorizing them into filtering methods, embedding methods, and wrapping methods. It includes details such as the number of features selected, the detection rate, and the false alarm rate for each method on each dataset. This table serves as a comprehensive overview of how these methods perform in the context of feature selection for intrusion detection.

## Filtered feature selection method

Filtered feature selection algorithms do not use explicit criteria to determine the size of the subset. Instead, they rank features based on various evaluation metrics and select the top N features with the highest scores. This selection process is based on the intrinsic characteristics of the dataset and does not consider feedback from classification results for the features already selected. By focusing on feature ranking and selection independently of the classification model, filtered feature selection algorithms aim to identify the most relevant features for the given dataset without being influenced by the performance of a specific classifier. *Amiri et al. (2011)* introduced a mutual information-based feature selection (MIFS) technique for NIDS. However, the accuracy of mutual information estimation may be compromised in scenarios with limited data, resulting in the identification of suboptimal sets of features. *Ambusaidi et al. (2016)* proposed a mutual information-based method to select optimal feature subset for classification from linear and nonlinear correlated data. The ARM feature selection model proposed by *Moustafa & Slay (2017)* focuses on enhancing detection performance by filtering out irrelevant features, retaining only significant ones, and leveraging association rule mining to identify feature

**Table 1 Summary of related works.**

| | Method | Dataset | Num of selection/ total | Detection rate | False alarm rate | Weaknesses |
|---|---|---|---|---|---|---|
| Filter method | MIFS | NSLKDD | 20/41 | 0.934 | – | Redundancy between features not considered |
| | IG | NSLKDD | 8/41 | 0.886 | 0.117 | The accuracy of mutual information is strongly influenced by the quality of the data |
| | ARM | NSLKDD | 11/41 | 0.997 | 0.003 | Overexploitation of high detection rate and neglecting the impact of false alarm rate on NIDS |
| Embedded method | PCA+EFC | CICIDS2017 | – | 0.957 | – | Highly influenced by noise data |
| | RFA | ISCX2012 | – | 0.896 | 0.026 | Redundancy in the selected subset of features |
| | XGBoost | UNSWNB15 | 19/43 | 0.908 | – | Due to the randomness of XGBoost, the calculation results of feature importance will fluctuate to a certain extent |
| Wrapper method | PSO | NSLKDD | 37/41 | 0.637 | 0.03 | Only combining PSO, GA and ACO does not make any improvement |
| | | UNSWNB15 | 19/43 | 0.863 | 0.037 | |
| | WOA | CICIDS2017 | – | 0.959 | – | Combine genetic operators to improve the search space, sacrificing the convergence speed |
| | SPIO | NSLKDD | 18/41 | 0.817 | 0.064 | No improvements to the PIO algorithm itself |
| | | UNSWNB15 | 14/43 | 0.897 | 0.052 | |
| | CPIO | NSLKDD | 5/41 | 0.866 | 0.088 | Only change the position mapping method of SPIO |
| | | UNSWNB15 | 5/43 | 0.894 | 0.034 | |
| | CCIHBO | NSLKDD | – | 0.971 | – | Introducing levy flight and LE-HBO, the algorithm complexity is too high |
| | | WUSTLIIOT | – | 0.976 | – | |
| | OWSA | NSLKDD | – | 0.9938 | – | Selecting parameters individually for each dataset increases the complexity of the model |
| | | CICIDS2017 | – | 0.97 | – | |
| | DBDE-QDA | NSLKDD | – | 0.854 | 0.150 | Excessive pursuit of reducing NIDS classification calculation time leads to a decrease in detection rate |
| | | UNSWNB15 | – | 0.768 | 0.057 | |

combinations with strong correlations. The comprehensive results show that ARM effectively minimizes false alarms and significantly reduces processing time while maintaining accuracy. *Stiawan et al. (2020)* conducted experiments using the mutual information selection technique with a NIDS on 20% of the streams from the CIC-IDS-2017 dataset. By reducing the number of features selected, the accuracy decreased, but the execution time also decreased significantly.

## Embedded feature selection method

Embedded feature selection algorithms are integrated with the machine learning model training process in a seamless manner. This approach offers the advantage of performing feature selection and model training simultaneously, resulting in optimized performance in both aspects. Embedded feature selection algorithms view the feature selection process as an integral part of model training. Feature weights are assigned concurrently with model training, all within a unified framework. *Yulianto, Sukarno & Suwastika (2019)* sought to enhance machine learning-based NIDS by incorporating principal component analysis and ensemble feature selection techniques for feature selection. Due to AdaBoost's high

sensitivity to abnormal data, combining PCA and EFS methods failed to achieve significant accuracy levels. *Hamed, Dara & Kremer (2018)* introduced a feature selection method to improve NIDS using recursive feature addition (RFA) and bipartite graph techniques. However, RFA neglects the redundancy and correlation between features, resulting in redundancy in the selected subset of features. *Kasongo & Sun (2020)* proposed integrating XGBoost into NIDS for feature selection and analyzed its effectiveness on the UNSW-NB15 dataset. According to XGBoost's feature importance ranking, 19 out of 43 features were identified as crucial. Using XGBoost's feature selection method in binary classification improved accuracy by 1.9%.

## Wrapper feature selection method

Wrapper feature selection algorithm approaches feature selection as a search problem, following a two-step process of identifying the most suitable feature set and assessing these chosen features. This cycle continues iteratively until specific termination criteria are satisfied. *Tama, Comuzzi & Rhee (2019)* introduced an enhanced NIDS that incorporates a hybrid feature selection approach. The approach including the Ant Colony Algorithm, particle swarm optimization, and the Genetic Algorithm to reduce feature size. Additionally, they proposed a two-stage classifier with Rotating Forest and Bagging methods. The model achieved an accuracy of 85.8% for the NSL-KDD dataset and 91.27% for the UNSW-NB15 dataset. *Vijayanand & Devaraj (2020)* proposed an enhanced methods that combines the Whale Optimization Algorithm (WOA) with genetic algorithm operators to prevent the convergence to local optimal solutions. By broadening the search space of the WOA, the approach aims to improve intrusion detection by extracting valuable features from network data. *Alazzam, Sharieh & Sabri (2020)* proposed two improved PIO algorithms, SPIO and CPIO. They conducted comparative experimental assessments to measure accuracy and false alarm rates, aiming to increase iteration speed without compromising precision. However, they did not make any improvements to the fundamental swarm intelligence algorithm, which limited the exploration of feature subsets. This constraint arises from the flock's inclination towards a greedy flight strategy that prioritizes superior positions exclusively. *Ye et al. (2023)* proposed an enhanced Collaborative Evolutionary HBO (CCIHBO) algorithm, which refines the conventional HBO method by incorporating Levy flight and Elite Opposition Learning strategy (LE-HBO). This augmentation aims to enhance the algorithm's efficacy in optimising solutions, resulting in a notable enhancement of nearly 15% in performance on extensive intrusion detection datasets like NSL-KDD, WUSTL-IIOT, and HAId. *Aldabash & Akay (2024)* proposed the Optimal Whale Sine Algorithm (OWSA) for selecting relevant features, leveraging the Sine Cosine Algorithm (SCA) optimization process. They further proposed the fusion of SCA with the Whale Optimization Algorithm (WOA) to address their respective limitations through hybridization. The experimental findings indicate that when combined with the Artificial Neural Network Weighted Random Forest (AWRF), the OWSA achieved an accuracy of 99.92% on the NSL-KDD dataset and 98% on the CICIDS2017 dataset. *Zorarpaci (2024)* presented a rapid wrapper feature selection technique, termed DBDE-QDA, which integrates two-class binary

differential evolution (DBDE) and quadratic discriminant analysis (QDA) to accelerate the process of wrapper feature selection. This approach aims to swiftly identify the optimal prediction features with minimal dimensions, thereby reducing the computational time needed. The experimental results demonstrate that DBDE-QDA offers decreased computational costs and effectively shortens the classification algorithm's computational time for network intrusion detection systems (NIDS). However, it may lead to a slight reduction in the detection rate for certain intrusion detection dataset.

In conclusion, while existing feature selection algorithms have partially addressed the challenges of intrusion detection systems (IDS) in 5G network environments, they still suffer from issues such as excessive feature selection, disregard for feature correlations, sensitivity to abnormal traffic, and difficulty in processing large-scale data. These challenges can reduce the model's generalisation ability, increase complexity, and decrease stability, thereby affecting detection performance and efficiency. To address these challenges, this article proposes a feature selection method based on an improved binary pigeon swarm optimization algorithm. In comparison to existing methods, the proposed approach incorporates mutation and simulated annealing mechanisms in the map and compass operator stages. These modifications are designed to enhance population diversity, expand the search space, and facilitate the acceptance of new solutions that are worse than the current solution with a certain probability, thereby facilitating escape from local optima. Furthermore, the proposed algorithm incorporates a population attenuation factor in the landmark operator stage. This factor dynamically adjusts the population size of each iteration based on the fitness distribution of the population, thus controlling the algorithm's convergence speed. The objective is to achieve improvements in key performance indicators such as detection rate, false alarm rate, and processing time.

## CONTINUOUS PIGEON INSPIRED OPTIMIZER

In 2014, *Duan & Qiao (2014)* researched pigeon behavior. They found that pigeons use geomagnetic cues and landmarks to navigate, determine direction, and find their nests. Based on these findings, the PIO algorithm was developed to imitate pigeons' migration behaviors and help find optimal solutions through communication and cooperation. The algorithm includes the map and compass operator phase and the landmark operator phase.

### Map and compass operator phase

The map and compass operator phase emulates how the sun and geomagnetic forces influence pigeon navigation. Pigeons assess the sun's position and geomagnetic cues to make real-time adjustments to their flight direction and strategize optimal routes. As pigeons approach their destination, they rely less on solar and geomagnetic guidance. During this phase, each pigeon is characterized by its positional and velocity data.

The PIO algorithm defines $V_i^t$ as the velocity of the ($i$)-th pigeon in the ($t$)-th iteration, and $P_i^t$ as its position. In each iteration, every pigeon adjusts its position $P_i^t$ and velocity $V_i^t$ according to Eqs. (1) and (2) (*Duan & Qiao, 2014*):

$$V_i^t = V_i^{t-1} e^{-Rt} + rand\left(P_{global} - P_i^{t-1}\right) \tag{1}$$

$$P_i^t = P_i^{t-1} + V_i^{t-1} \tag{2}$$

In Eq. (1), $R$ represents the map and compass operator, $t$ denotes the current number of iterations and the random function $rand \in [0, 1]$, $P_{global}$ stands for the globally optimal position obtained by comparing the positions of all pigeons in $(t-1)$-th iteration.

## Landmark operator phase

The landmark operator mimics how navigational landmarks affect pigeons. Pigeons have the ability to rapidly store details about surrounding landmarks during navigation. As they approach the target location, pigeons rely on nearby landmarks to construct a mental map and fine-tune their position and speed in response to these landmarks until they reach the intended destination. If a pigeon is unfamiliar with the local landmark, it will adjust its flight based on the flight patterns of nearby pigeons that are familiar with the landmark. During the iterative process of the landmark operator phase, pigeons are eliminated based on their fitness disparity, removing the less adapted half of the pigeons. The central position of the remaining, more adept pigeons is then computed as the reference direction within the population. The position of the pigeon is updated at this phase based on Eqs. (3)–(5) (*Duan & Qiao, 2014*).

$$P_{center}^{t-1} = \frac{\sum\limits_{i=1}^{Num_{pigeon}^{t-1}} P_i^{t-1} \cdot Fitness\left(P_i^{t-1}\right)}{Num_{pigeon}^{t-1} \cdot \sum\limits_{i=1}^{Num_{pigeon}^{t-1}} Fitness\left(P_i^{t-1}\right)} \tag{3}$$

The iteration of the center position of the pigeon group can be denoted by Eq. (3), where $Num_{pigeon}^t$ denotes the quantity of pigeon groups in the $(t)$-th iteration, $t$ signifies the present iteration number, and the fitness function *Fitness* adopts distinct valuation methods for various issues. In instances where the aim is to minimize a problem, involving the reciprocal, $P_{center}^{t-1}$ represents the position of the pigeon center (desired destination) in the $(t-1)$-th iteration.

$$Num_{pigeon}^t = \frac{sort\left(Num_{pigeon}^{t-1}\right)}{2} \tag{4}$$

Among them, the sorting function *sort* represents sorting the pigeon group according to adaptability. The iterative decay of the population can be described by Eq. (4).

$$P_i^t = P_i^{t-1} + rand\left(P_{center}^{t-1} - P_i^{t-1}\right) \tag{5}$$

Equation (5) describes how the remaining flock adjusts its position relative to the center position of the flock by incorporating the random function $rand \in [0, 1]$.

The PIO algorithm is logically coherent, easy to understand, robust, and has significant research implications. The PIO algorithm has been shown to be effective in addressing various challenges, including the unmanned aerial vehicle path planning dilemma

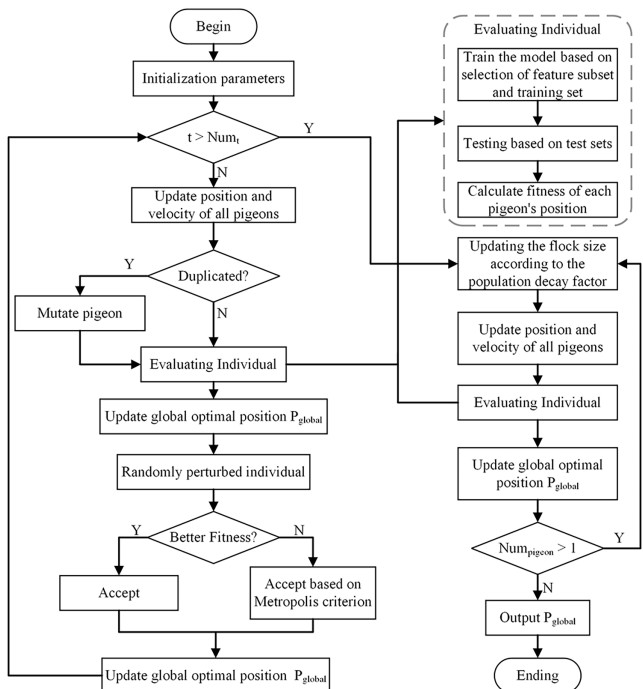

**Figure 1 Proposed SABPIO method.**

(*Yuan & Duan, 2024*), the security concerns associated with medical image encryption (*Geetha et al., 2022*), and the optimization of large-scale hydroelectric short-term generation (*Tian et al., 2020*). While the PIO algorithm exhibits superior performance compared to other population intelligence algorithms, it still suffers from drawbacks such as rapid convergence and a tendency to explore. To address the issue of rapid iteration and susceptibility to local optima in the PIO algorithm, this study introduces mutation and simulated annealing techniques to broaden the search scope. Additionally, a population decay factor is suggested to regulate the algorithm's convergence rate, thereby enhancing its overall performance, diminishing feature selection data dimensionality, and boosting the efficiency of intrusion detection.

## PROPOSED IMPROVEMENT OF PIO

This article proposes a method to integrate the Simulated Annealing into the Binary PIO (SABPIO) algorithm for feature selection in NIDS. The approach incorporates simulated annealing and mutation into the conventional PIO algorithm, expanding the search scope and mitigating the risk of local optima. Additionally, a population decay factor is introduced to regulate the algorithm's convergence speed. The proposed SABPIO feature selection algorithm is shown in Fig. 1.

The proposed method generates the initial positions of the pigeons by utilizing randomly chosen features from the dataset, establishing the initial population. Decision tree (DT) and Random Forest (RF) classifiers are used to determine the search subject, which is the position of the pigeon closest to the target. These classifiers evaluate the fitness of each pigeon position within the population, and the positions of the remaining pigeons

| Feature Dimension | 1 | 2 | 3 | 4 | ... | d-1 | d |
|---|---|---|---|---|---|---|---|
| P | 0 | 0 | 1 | 1 | ... | 0 | 0 |

**Figure 2  Location encoding for pigeon.**

are adjusted based on the optimal solution. Following this, the pigeon swarm undergoes probabilistic positional adjustments utilizing simulated annealing. This mechanism aids in steering clear of local optimal solutions and enhances solution diversity within the search process. Finally, the population attenuation factor is used to decrease the pigeon population, which improves the exploration of solutions within the search space. The output of each iteration serves as the input for the following iterations until the optimal feature subset is identified.

## Pigeon encoding

The pigeon position symbolizes the potential selection for features, with a single pigeon representing a particular feature subset. As shown in Fig. 2, the upper vector in the encoding denotes the feature's order number (dimension), while the lower vector indicates the pigeon's binary position within each dimension. The spatial dimension denoted by $d$ explored by the pigeon corresponds to the quantity of network features. In the binary vector $P_i = (p_{i1}, p_{i2}, \ldots, p_{id})$, when $p_{ij} = 1$, it signifies that feature $j$ within the feature subset represented by pigeon $i$ is chosen. Conversely, when $p_{ij} = 0$, it indicates that feature $j$ in the feature subset represented by pigeon $i$ is not selected, meaning it is excluded from the optimal feature subset.

## Fitness function

The Fitness Function evaluates the fitness of every individual. It is formulated considering the individual's traits and the specifications of the given problem, converting the individual into a numerical value that reflects their suitability for problem-solving. Given that the two metrics of true positive rate (TPR) and false negative rate (FPR) serve as effective gauges for assessing the model's efficacy in identifying attacks and managing false positives in routine activities, a majority of researchers opt to employ TPR and FPR (*Thakkar & Lohiya, 2023*) as the fitness criteria (*Louk & Tama, 2023*) in their calculations.

$$TPR = \frac{TP}{TP + FN} \tag{6}$$

$$FPR = \frac{FP}{TN + FP} \tag{7}$$

Equations (6) and (7) provide the calculation formulas for TPR and FPR. In the feature selection problem of a NIDS, TP refers to the system identifying abnormal traffic as attack events, and TN refers to the system identifying normal traffic as non-attack events. FP refers to the system identifying correct traffic as an attack event, and FN refers to the system identifying abnormal traffic as a non-attack event. The SABPIO algorithm incorporates the ratio of selected features into the fitness function to account for their potential impact on intrusion detection time. This adjustment aims to eliminate features

within the subset that do not significantly contribute to detection accuracy. The present study also introduces a fitness function formula, shown in Eq. (8), which reframes the optimization of feature selection as a minimization task.

$$Fitness = \frac{1 + \lambda * Num_{SF}}{TPR + 1 - FPR} \tag{8}$$

where $Num_{SF}$ denotes the number of selected features, $\lambda$ is the weighting factor and $\lambda \in (0, 1)$. In Eq. (8), the numerator considers the impact of the selected feature quantity on adaptability, while the denominator accounts for the NIDS performance's influence on adaptability. Through the Fitness Function, the SABPIO algorithm strikes a balance between feature quantity and classification performance, effectively enhancing classification efficiency while ensuring the accuracy of NIDS detection.

## Binary mapping strategy

The continuous pigeon-inspired optimizer (CPIO) algorithm involves a process of continual spatial repositioning for the pigeon, enabling it to traverse any point within space. However, in certain discrete scenarios such as feature selection, the pigeon's position, representing a solution matrix, consists of binary values of 0 and 1. Therefore, updating continuous values requires the application of appropriate position adjustment techniques in addition to discretization operations.

In the context of feature selection, the pigeon's position within each dimension of the search space is constrained to 0 or 1. However, the velocity associated with each dimension is not subject to such limitations. Therefore, the integration of a conversion function becomes essential to effectively map the position variables onto binary values. After conducting experimental analysis, we selected the Tanh function to map pigeon velocities into the binary space. The Tanh function formula (*Sood et al., 2023*) is shown in Eq. (9), and the positions of the individual pigeon flocks are updated using a uniform random number $r \in (0, 1)$, with the Tanh value through Eq. (10) in this article.

$$Tanh(x) = \frac{2}{1 - e^{-2x}} - 1 \tag{9}$$

$$P_i^t[j] = \begin{cases} P_i^{t-1}[j] & Tanh\left(V_i^t[j]\right) > r \\ -P_i^{t-1}[j] & Tanh\left(V_i^t[j]\right) < -r \\ P_{global}^{t-1}[j] & \text{otherwise} \end{cases} \tag{10}$$

The individual pigeon's position is updated according to Eq. (10). In this process, for each dimension of the position, the velocity is evaluated against a randomly generated number $r$ and the current dimension's pigeon velocity. In instances where the mapping value $Tanh\left(V_i^t[j]\right) > r$ in the ongoing velocity iteration, there is a strong positive correlation between velocity and position, the position from the previous iteration in the current dimension is preserved. If $Tanh\left(V_i^t[j]\right) < -r$, there is a strong negative correlation between velocity and position, a reverse operation is applied to the position from the prior iteration in the current dimension. In all other scenarios where there is a weak correlation

between velocity and position, the optimal position value from the previous iteration is directly utilized.

## Improved map and compass operator phase

(1) Simulated annealing

To tackle the issue of rapid convergence observed in conventional PIO algorithms, the proposed approach introduces a simulated annealing mechanism during this phase to prevent premature trapping in local optimal solutions. In the map and compass operator phase, each iteration of the pigeon undergoes adjustments to both velocity and position. The influence of the map and compass operators $R$ on the population decreases as the algorithm approaches later stages of iteration. At this point, the algorithm relies mainly on the current globally optimal position $P_{global}$.

This approach integrates simulated annealing to enhance the inner loop with each iteration. During the loop, a random pigeon undergoes perturbation, resulting in the modification of one value in the vector, such as changing a 1-value to a 0-value. Then, the fitness is recalculated, and a new feature subset is accepted based on the probability determined by the Metropolis criterion (*Hao et al., 2023*). The purpose of this criterion is to determine whether to accept a new state based on the change in energy value before and after the state modification. The study employs the Metropolis criterion outlined in Eq. (11):

$$p(P_{global} \Rightarrow P_i') = \begin{cases} 1 & \Delta E < 0 \\ e^{-\frac{\Delta E}{T}} & \text{otherwise} \end{cases} \tag{11}$$

where $p(P_{global} \Rightarrow P_i')$ represents the probability of accepting the new solution $P_i'$, while $\Delta E$ represents the energy difference, denoted as *Fitness* $(P_i')$ − *Fitness* $(P_{global})$ in this context. In this article, the fitness is transformed into a minimization problem. If the fitness of the new solution $P_i'$ is lower than the fitness of the globally optimal solution $P_{global}$, implies that the feature subset represented by the new solution $P_i'$ is superior to the globally optimal solution $P_{global}$, $P_i'$ is accepted as the current globally optimal solution with a probability of 1. Conversely, the probability $p(P_{global} \Rightarrow P_i')$ is used to determine whether the new solution should be accepted.

(2) Multi-dimensional similarity strategy

During the map and compass operator phase, the white pigeon adjusts its flight position by tracking the position of the best pigeon (blue pigeon), as shown in Fig. 3.

Instead, the pigeon computes its velocity by subtracting its own position vector from the global optimal vector. In a discrete problem, it is not feasible to directly subtract the pigeon's position vector as done in the continuous problem due to the nature of discrete variables. This article introduces a multi-dimensional similarity strategy for computing pigeon velocities. The strategy includes metrics such as Pearson correlation coefficient (*Saviour & Samiappan, 2023*), cosine similarity (*Alazzam, Sharieh & Sabri, 2020*), and Jaccard similarity coefficient (*Yin et al., 2023*), as shown in Eqs. (12)–(14). All three similarity indicators have limitations. To balance these limitations, this article employs
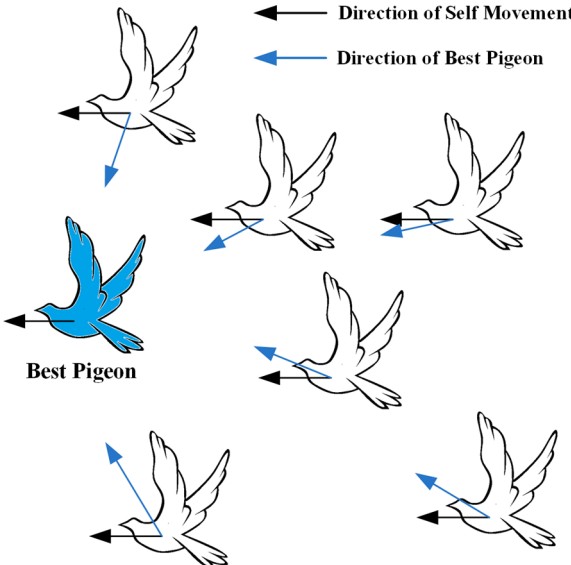

**Figure 3 White pigeons adjusts its flight position according to the map and the compass operator.** The silhouette of the flying pigeon in the image is sourced from (freeimages.com) (https://www.freeimages.com/cn/vector/flying-dove-clip-art-4757856?ref=vectorhq).

weighted calculations to avoid relying on a single indicator. In this phase, each pigeon updates its velocity and position for this iteration based on Eqs. (15) and (10).

$$Pearson\ (P_i, P_{global}) = \frac{cov\ (P_i, P_{global})}{\sqrt{var\ (P_i) \cdot var\ (P_{global})}} \tag{12}$$

$$Consine\ (P_i, P_{global}) = \frac{P_i \cdot P_{global}}{||P_i|| \cdot ||P_{global}||} \tag{13}$$

$$Jaccard\ (P_i, P_{global}) = \frac{|P_i \cap P_{global}|}{|P_i \cup P_{global}|} \tag{14}$$

$$V_i = CompositeSim\ (P_i, P_{global})$$
$$= \omega_1\ Pearson\ (P_i, P_{global}) + \omega_2\ Consine\ (P_i, P_{global}) + \omega_3\ [2\ Jaccard\ (P_i, P_{global}) - 1] \tag{15}$$

Equation (15) requires normalizing Jaccard's correlation coefficient to the range of [−1, 1], given that Pearson's correlation coefficient and Cosine similarity have a value range of [−1, 1], and Jaccard's similarity coefficient ranges from [0, 1]. In this, $\omega_1$, $\omega_2$ and $\omega_3$ represent the weighting coefficients of Pearson's correlation coefficient, cosine similarity, and Jaccard's similarity coefficient, respectively, with $\omega_1 + \omega_2 + \omega_3 = 1$.

(3) Mechanism of mutation

When the initial number of pigeon flocks is high, there is a greater chance that two pigeons will represent the same solution, which will reduce the search ability of the algorithm. Therefore, this approach includes a mutation mechanism in the flock's position updates. It checks for the existence of a solution with a similar position before adding the updated pigeon to the flock. If such a solution is found, all dimensions of the current

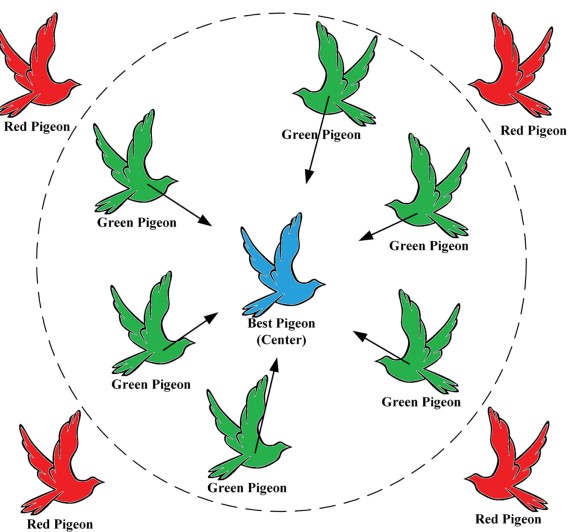

**Figure 4 Elimination of red pigeon by adaptation calculation, green pigeon flying towards desired destination (blue pigeon).** Silhouette of flying pigeon from (freeimages.com) (https://www.freeimages.com/cn/vector/flying-dove-clip-art-4757856?ref=vectorhq).

pigeon undergo mutation based on the probability derived from a uniformly distributed random number $r \in [0, 1]$, which expands the search space.

### Improved landmark operator phase

During each iteration of the landmark operator phase, the pigeons are sorted based on their fitness values. Then, half of the pigeons with lower fitness values are eliminated. The current center position of the remaining dominant breeder flock is considered the desired destination of the flock. The remaining flock adjusts its flight position towards the position of the desired destination, also known as the Blue Pigeon, as shown in Fig. 4.

The population decay factor $\alpha$ was proposed to regulate the decay rate of the population, as the traditional pigeonholing algorithm tends to converge too quickly and fall into local optimal solutions during the landmark operator phase.

$$\alpha = \beta \cdot e^{-\frac{t}{\log_2 Num_{pigeon}}} \tag{16}$$

$$Num_{pigeon}^{t} = \alpha \cdot sort\left(Num_{pigeon}^{t-1}\right) \tag{17}$$

Equation (16) defines $\beta$ as a constant between $(0, 1)$, and $t$ as the number of iterations of the landmark operator. The SABPIO algorithm improves the traditional pigeon colony algorithm by halving it with the number of iterations and updating the population size according to Eq. (17), thus prevent rapid population decay in the early stage. During the map and compass operator phase, all pigeons calculate their speed and position using Eqs. (15) and (10).

Algorithm 1 outlines the procedure for the feature selection algorithm SABPIO, which is based on an improved binary PIO framework. The upgraded algorithm introduces a simulated annealing loop within the initial phase of the main iteration loop to extend the

---

**Algorithm 1   Simulated Annealing Binary Pigeon Inspired Optimizer (SABPIO).**

**Input:** Number of pigeons $Num_{pigeon}$, Number of iterations $Num_t$, Fitness function $Fitness$, Number of annealing iterations $Num_{at}$

**Output:** Global optimal solution $P_{global}$

01: Randomly initialize velocity $V_i$ and position $P_i$ of all pigeons

02: for $t = 1$ to $Num_t$ do // Map and compass operator phase

03:     Update velocity $V_i^t$ and position $P_i^t$ for each pigeon by Eqs. (15) and (10)

04:     Check for duplicate items at each pigeon's position $[P_1, P_2.......,P_{Num_{pigeon}}]$

05:     Calculate fitness $Fitness(P_i)$ of each pigeon's position $[P_1, P_2.......,P_{Num_{pigeon}}]$

06:     Find global optimal solution $P_{global} = \min\{Fitness(P_i) \mid i \in [0, Num_{pigeon}]\}$

07:     Update global optimal solution $P_{global}$

08:     for $t = 1$ to $Num_t$ do

09:         Randomly perturbed individuals $P_i'$ and calculate fitness $Fitness(P_i')$

10:         Accept the new global optimal solution by Eq. (11)

11:     end for

12: end for

13: while $(Num_{pigeon} \geq 1)$ // Landmark operator phase

14:     Update center position of all pigeons $P_{center}$ by Eq. (3)

15:     Update number of pigeons $Num_{pigeon}$ by Eq. (17)

16:     Update velocity $V_i^t$ and position $P_i^t$ for each pigeon by Eqs. (15) and (10)

17:     Update global optimal solution $P_{global}$

18: end while

19: return $P_{global}$

---

exploration of the global search space. Additionally, the secondary while loop uses a population decay factor to regulate the pace of population reduction and mitigate premature convergence of the algorithm.

# EXPERIMENTS AND RESULTS

## Experimental dataset

### (1) UNSW-NB15 dataset

The UNSW-NB15 dataset represents network traffic data collected by a cybersecurity research laboratory in Australia utilizing the IXIA Perfect Storm tool. It comprises four CSV files encompassing 254,047 data entries, featuring nine attack classifications, 43 descriptive attributes, and two classification labels for each entry. The detailed feature attributes of this dataset are outlined in Table 2.

### (2) NSL-KDD dataset

The NSL-KDD dataset serves as an updated iteration of the renowned KDD99 dataset, comprising 148,517 entries. Each entry is composed of 41 descriptive attributes and one class label, encompassing a total of 39 attack classifications. Within the training set, there

**Table 2 UNSW-NB15 dataset features and types.**

| No | Feature | Type | No | Feature | Type |
|---|---|---|---|---|---|
| 1 | id | Integer | 24 | dwin | Integer |
| 2 | dur | Float | 25 | tcprtt | Float |
| 3 | proto | Nominal | 26 | synack | Float |
| 4 | service | Nominal | 27 | ackdat | Float |
| 5 | state | Nominal | 28 | smean | Integer |
| 6 | spkts | Integer | 29 | dmean | Integer |
| 7 | dpkts | Integer | 30 | trans_depth | Integer |
| 8 | sbytes | Integer | 31 | response_body_len | Integer |
| 9 | dbytes | Integer | 32 | ct_srv_src | Integer |
| 10 | rate | Float | 33 | ct_state_ttl | Integer |
| 11 | sttl | Integer | 34 | ct_dst_ltm | Integer |
| 12 | dttl | Integer | 35 | ct_src_dport_ltm | Integer |
| 13 | sload | Float | 36 | ct_dst_sport_ltm | Integer |
| 14 | dload | Float | 37 | ct_dst_src_ltm | Integer |
| 15 | sloss | Integer | 38 | is_ftp_login | Binary |
| 16 | dloss | Integer | 39 | ct_ftp_cmd | Integer |
| 17 | sinpkt | Float | 40 | ct_flw_http_mthd | Integer |
| 18 | dinpkt | Float | 41 | ct_src_ltm | Integer |
| 19 | sjit | Float | 42 | ct_srv_dst | Integer |
| 20 | djit | Float | 43 | is_sm_ips_ports | Binary |
| 21 | swin | Integer | 44 | attack_cat | Nominal |
| 22 | stcpb | Integer | 45 | label | Binary |
| 23 | dtcpb | Integer | | | |

are 125,973 data points encompassing 22 distinct attack types, while the test set consists of 22,544 entries featuring a further 17 attack categories. The defining attributes within this dataset are detailed in Table 3.

(3) CIC-IDS-2017 dataset

The CIC-IDS-2017 dataset encompasses network traffic data gathered by the Canadian Institute of Cybersecurity (CIC) from authentic network scenarios. This dataset is constructed using actual network traffic captures, encompassing a broad spectrum of network intrusions and regular activities. It comprises real network traffic observed across various laboratory network environments designed to replicate the network traits found in commercial and industrial entities. The CIC-IDS-2017 dataset encompasses a diverse array of network intrusion behaviors and standard network operations. It aligns with the traits of contemporary networks and stands as one of the presently recommended datasets. Featuring 2,830,743 records, each entry comprises 78 defining attributes and one class label, covering a total of eight attack classifications. The detailed characteristics of the dataset are presented in Table 4.

**Table 3 NSL-KDD dataset features and types.**

| No | Feature | Type | No | Feature | Type |
|----|---------|------|----|---------|------|
| 1 | duration | Float | 22 | is_guest_login | Integer |
| 2 | protocol_type | Integer | 23 | count | Float |
| 3 | service | Integer | 24 | srv_count | Float |
| 4 | flag | Integer | 25 | serror_rate | Float |
| 5 | src_bytes | Float | 26 | srv_serror_rate | Float |
| 6 | dst_bytes | Float | 27 | rerror_rate | Float |
| 7 | land | Integer | 28 | srv_rerror_rate | Float |
| 8 | wrong_fragment | Float | 29 | same_srv_rate | Float |
| 9 | urgent | Float | 30 | diff_srv_rate | Float |
| 10 | hot | Float | 31 | srv_diff_host_rate | Float |
| 11 | num_failed_logins | Float | 32 | dst_host_count | Float |
| 12 | logged_in | Float | 33 | dst_host_srv_count | Float |
| 13 | num_compromised | float | 34 | dst_host_same_srv_rate | Float |
| 14 | root_shell | Float | 35 | dst_host_diff_srv_rate | Float |
| 15 | su_attempted | Float | 36 | dst_host_same_src_port_rate | Float |
| 16 | num_root | Float | 37 | dst_host_srv_diff_host_rate | Float |
| 17 | num_file_creations | Float | 38 | dst_host_serror_rate | Float |
| 18 | num_shells | Float | 39 | dst_host_srv_serror_rate | Float |
| 19 | num_access_files | Float | 40 | dst_host_rerror_rate | Float |
| 20 | num_outbound_cmds | Float | 41 | dst_host_srv_rerror_rate | Float |
| 21 | is_host_login | Integer | 42 | Class | Integer |

**Table 4 CIC-IDS-2017 dataset features and types.**

| No | Feature | Type | No | Feature | Type | No | Feature | Type |
|----|---------|------|----|---------|------|----|---------|------|
| 1 | Destination port | Integer | 28 | Bwd IAT std | Float | 55 | Avg bwd segment size | Float |
| 2 | Flow duration | Integer | 29 | Bwd IAT max | Integer | 56 | Fwd header length | Integer |
| 3 | Total fwd packets | Integer | 30 | Bwd IAT min | Integer | 57 | Fwd avg bytes/bulk | Float |
| 4 | Total backward packets | Integer | 31 | Fwd PSH flags | Integer | 58 | Fwd avg packets/bulk | Float |
| 5 | Total length of fwd packets | Integer | 32 | Bwd PSH flags | Integer | 59 | Fwd avg bulk rate | Float |
| 6 | Total length of bwd packets | Integer | 33 | Fwd URG flags | Integer | 60 | Bwd avg bytes/bulk | Float |
| 7 | Fwd packet length max | Integer | 34 | Bwd URG flags | Integer | 61 | Bwd avg packets/bulk | Float |
| 8 | Fwd packet length min | Integer | 35 | Fwd header length | Integer | 62 | Bwd avg bulk rate | Float |
| 9 | Fwd packet length mean | Float | 36 | Bwd header length | Integer | 63 | Subflow fwd packets | Integer |
| 10 | Fwd packet length std | Float | 37 | Fwd packets/s | Float | 64 | Subflow fwd bytes | Integer |
| 11 | Bwd packet length max | Integer | 38 | Bwd packets/s | Float | 65 | Subflow bwd packets | Integer |
| 12 | Bwd packet length min | Integer | 39 | Min packet length | Integer | 66 | Subflow bwd bytes | Integer |
| 13 | Bwd packet length mean | Float | 40 | Max packet length | Integer | 67 | Init_Win_bytes_forward | Integer |
| 14 | Bwd packet length std | Float | 41 | Packet length mean | Float | 68 | Init_Win_bytes_backward | Integer |
| 15 | Flow bytes/s | Float | 42 | Packet length Std | Float | 69 | Act_data_pkt_fwd | Integer |
| 16 | Flow packets/s | Float | 43 | Packet length variance | Float | 70 | Min_seg_size_forward | Integer |

(Continued)

| No | Feature | Type | No | Feature | Type | No | Feature | Type |
|----|---------|------|----|---------|------|----|---------|------|
| 17 | Flow IAT mean | Float | 44 | FIN flag count | Integer | 71 | Active mean | Float |
| 18 | Flow IAT std | Float | 45 | SYN flag count | Integer | 72 | Active std | Float |
| 19 | Flow IAT max | Integer | 46 | RST flag count | Integer | 73 | Active max | Integer |
| 20 | Flow IAT min | Integer | 47 | PSH flag count | Integer | 74 | Active min | Integer |
| 21 | Fwd IAT total | Integer | 48 | ACK flag count | Integer | 75 | Idle mean | Float |
| 22 | Fwd IAT mean | Float | 49 | URG flag count | Integer | 76 | Idle std | Float |
| 23 | Fwd IAT std | Float | 50 | CWE flag count | Integer | 77 | Idle max | Integer |
| 24 | Fwd IAT max | Integer | 51 | ECE flag count | Integer | 78 | Idle min | Integer |
| 25 | Fwd IAT min | Integer | 52 | Down/up ratio | Integer | 79 | Label | Binary |
| 26 | Bwd IAT total | Integer | 53 | Average packet size | Float | | | |
| 27 | Bwd IAT mean | Float | 54 | Avg fwd segment size | Float | | | |

(4) Data preprocessing

1) Data conversion

During the algorithm's execution, solely numerical data is utilized for training and testing purposes. Hence, the initial step involves transforming non-numeric data within the dataset into numerical format. Taking the UNSW-NB15 dataset as a case in point, out of the 45 descriptive attributes, three are non-numeric and necessitate conversion *via* one-hot encoding. For instance, consider the "proto" attribute containing 133 distinct values such as "tcp," "udp," and "sctp." These values are encoded into numerical representations ranging from 0 to 132. Subsequently, the "service" and "state" attributes undergo a similar transformation into numerical format utilizing the aforementioned method.

2) Normalized

Normalization is a crucial data preprocessing technique that facilitates the comparison and analysis of data by standardizing data with varying scales and distributions onto a uniform scale (*Devendiran & Turukmane, 2024*). This process enhances the accuracy and efficiency of data analysis and machine learning algorithms while mitigating biases stemming from variations across different variables. The normalization formula, as illustrated in Eq. (18), plays a pivotal role in this standardization process.

$$x_{norm} = \frac{x - x_{min}}{x_{max} - x_{min}} \tag{18}$$

where $x_{max}$ is the maximum of the eigenvalues, $x_{min}$ is the minimum of the eigenvalues and $x_{norm}$ is the output value which is between [0, 1].

## Evaluation indicators

Multiple metrics are available for evaluating feature selection algorithms. In this study, we use evaluation metrics derived from the confusion matrix, including detection rate (DR), false alarm rate (FAR), Accuracy (Acc), Precision (Pre), and F1-score (*Thakkar & Lohiya, 2023*), as shown in Eqs. (19)–(23). Table 5 shows the confusion matrix.

**Table 5 Confusion matrix.**

| | | Predicted | |
|---|---|---|---|
| | | Positive | Negative |
| Actual | Positive | True positive (TP) | False negative (FN) |
| | Negative | False positive (FP) | True negative (TN) |

(1) Detection rate (DR)

The detection rate, also known as the true positive rate (TPR), signifies the capacity to accurately recognize all true positive samples. It represents the proportion of positive samples correctly identified by the model. In the realm of network intrusion detection, it signifies the percentage of intrusion events effectively identified by the model. A heightened detection rate implies the model's enhanced ability to accurately identify potential intrusions.

$$DR = TPR = \frac{TP}{TP + FN} \tag{19}$$

(2) False alarm rate (FAR)

The false alarm rate, denoted as the false positive rate (FPR), represents the ratio of negative samples that the model inaccurately identifies as positive samples. In the context of network intrusion detection, it signifies the percentage of normal behaviors erroneously identified as intrusions by the model. A diminished FPR indicates the model's efficacy in minimizing false alarms.

$$FAR = FPR = \frac{FP}{FP + TN} \tag{20}$$

(3) Accuracy

Accuracy refers to the proportion of correctly predicted samples by the model, serving as a measure of the model's overall predictive precision.

$$Accuracy = \frac{TP + TN}{TP + TN + FP + FN} \tag{21}$$

(4) Precision

Precision is the ratio of correctly identified positive samples by the model. In the context of network intrusion detection, it reflects the accuracy of the model in identifying all samples flagged as intrusions. Enhanced precision indicates greater reliability of the model in reporting alarms and a reduced occurrence of false alarms.

$$Precision = \frac{TP}{TP + FP} \tag{22}$$

(5) F1-score

F1-score is a measure that assesses the balance between precision and recall while taking into account the model's accuracy and comprehensiveness. It is calculated as the harmonic average of precision and recall.

$$\text{F1} - \text{Score} = 2 * \frac{Pre * DR}{Pre + DR} = \frac{2 * TP}{2 * TP + FP + FN} \tag{23}$$

## Experimental results

In this study, we conducted experiments to evaluate the proposed approach utilizing Python 3.11.2 on 64-bit Windows 11 operating system. The experiments were carried out on Intel(R) Core (TM) i5-11400H processor with 16.00 GB of RAM. All feature selection algorithms were assessed using the decision tree (DT) classifier and Random Forest (RF) classifier from the scikit-learn library for evaluation purposes. Compared to alternative base classifiers, DT and RF are less sensitive to missing values and more robust against outliers and noise. This makes them well-suited for assessing feature selection issues.

Table 6 delineates the parameter configurations of the SABPIO algorithm. Through rigorous experimental analysis, we found that the performance of the algorithm is optimal when the number of individuals in the pigeon swarm is within the range of [80, 150]. Consequently, we set the number of pigeons to 128. It's important to note that while a larger number of pigeons allows for a broader exploration of the search space, it also increases computational complexity. In the inner loop of the annealing iterations, only the new solution and the current local optimal solution are compared. As such, the number of iterations has a minimal impact on time complexity. Therefore, we set the number of iterations in the simulated annealing inner loop to 100. In the calculation of the fitness function, we considered the number of feature selections. If the weight factor is too large, the pigeon swarm may overly pursue feature subsets with fewer elements rather than optimal feature subsets. To prevent the fitness calculation from ignoring the impact of TPR and FPR, we set the weight factor of the number of feature selections to 0.0075.

The conducted experiments were based on preprocessed datasets comprising UNSW-NB15 (*Moustafa & Slay, 2015*), NSL-KDD (*Tavallaee et al., 2009*) and CIC-IDS-2017 (*Sharafaldin, Lashkari & Ghorbani, 2018*). The datasets were partitioned into training and testing sets using the train_test_split function from the scikit-learn library, with the stratify parameter ensuring stratified sampling based on the labels. The ratio of the training set to the testing set was 0.8 and 0.2, respectively. These experiments underwent 100 iterations employing the recommended feature selection methodology to pinpoint a compact feature set, with subsequent averaging of outcomes. The effectiveness of the suggested feature selection approach was appraised through juxtaposition with recognized feature selection techniques like CPIO (*Alazzam, Sharieh & Sabri, 2020*), SPIO (*Alazzam, Sharieh & Sabri, 2020*), XGBoost (*Kasongo & Sun, 2020*), PSO (*Tama, Comuzzi & Rhee, 2019*), ARM (*Moustafa & Slay, 2017*), IG (*Aljawarneh, Aldwairi & Yassein, 2018*), WOA (*Vijayanand & Devaraj, 2020*), and AdaBoost+EFS (*Yulianto, Sukarno & Suwastika, 2019*).

**Table 6  Detailed parameters of SABPIO.**

| Parameters | Value |
| --- | --- |
| Number of pigeons $Num_{pigeon}$ | 128 |
| Number of iterations $Num_t$ | 100 |
| Number of annealing iterations $Num_{at}$ | 100 |
| Weighting factor of Pearson $\omega_1$ | 0.4 |
| Weighting factor of Cosine $\omega_2$ | 0.4 |
| Weighting factor of Jaccard $\omega_3$ | 0.2 |
| Weighting factor of number of selections $\lambda$ | 0.0075 |
| Weighting factor of attenuation factor $\beta$ | 0.8 |

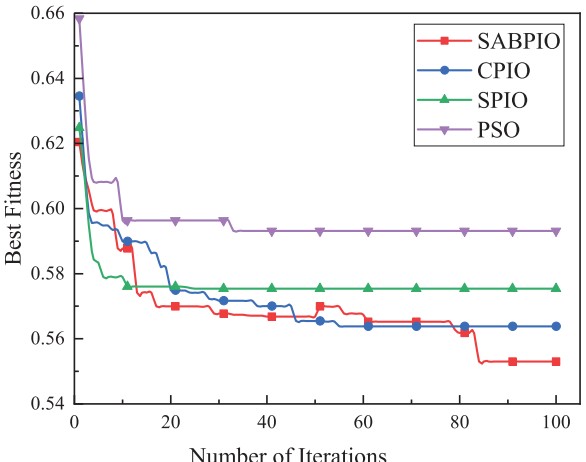

**Figure 5  Convergence curve on the UNSW-NB15 dataset by RF.**

(1) Results of UNSW-NB15

The performance, convergence, and efficiency of SABPIO were compared to those of CPIO, SPIO, XGBoost, PSO, and ARM algorithms using the UNSW-NB15 dataset. Figure 5 shows the convergence curves of SABPIO, CPIO, SPIO, and PSO during the feature selection process under the random forest classifier. In "Proposed Improvement of PIO" of the article, fitness is defined as a minimization problem. The data suggests that SABPIO converges faster than SPIO and PSO algorithms and achieves better fitness values with each iteration compared to CPIO, SPIO, and PSO algorithms.

In Fig. 5, it is evident that the SABPIO algorithm exhibits the swiftest rate of adaptation decay within the initial 30 iterations. Conversely, the SPIO algorithm showcases a rapid decay rate within the first 10 iterations; however, subsequent iterations reveal that the SPIO algorithm becomes ensnared in a locally optimal solution, impeding the exploration for a superior solution. By the 50th iteration, the SABPIO algorithm embraces a suboptimal solution based on the Metropolis criterion probability, resulting in a slight fitness increase. By the time all algorithms reach convergence at 100 iterations, it is apparent that the solution derived by SABPIO demonstrates reduced adaptation. The

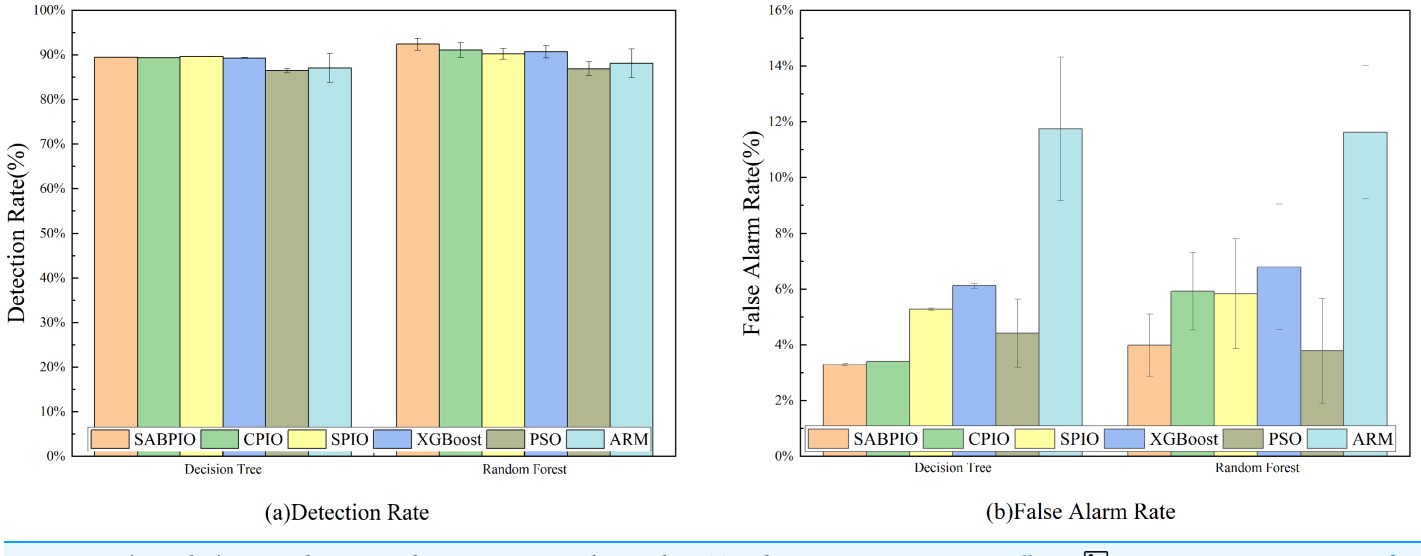

**Figure 6** (A and B) DR and FAR on the UNSW-NB15 dataset by DT and RF.

experimental findings affirm that SABPIO boasts enhanced convergence efficiency compared to SPIO and PSO, along with greater efficacy in the selected feature subset than CPIO, SPIO, and PSO.

In Fig. 6, the detection rate (DR) and false alarm rate (FAR) of the SABPIO algorithm, assessed on DT and RF classifiers with a subset of features selected by other algorithms, are presented. Each bar in the figure represents the results and standard deviation obtained from 100 repeated runs of the feature subset selected by the algorithm in the DT and RF classifiers. In Fig. 6A, it is observed that SPIO achieves the highest DR among the DT classifiers, slightly surpassing the performance of the SABPIO algorithm. However, it is crucial to acknowledge that DR is not the sole metric utilized in this study for evaluating the feature subset in network intrusion detection. Moving on to Fig. 6B, SABPIO exhibits a 2% lower FAR compared to SPIO, while only experiencing a marginal 0.2% reduction in DR. In comparison to CPIO, XGBoost, PSO, and ARM algorithms, the proposed SABPIO algorithm demonstrates advantages in both DR and FAR. Notably, the ARM algorithm prioritizes a high detection rate as the optimization objective, neglecting the impact of the false alarm rate on NIDS, thereby resulting in an elevated false alarm rate within this dataset. Within the RF classifiers, SABPIO showcases more significant improvements than other algorithms. The mean performance of the proposed algorithm across 100 repeated experiments surpasses that of other algorithms, with the standard deviation consistently maintained at a low level.

Figure 7 displays the accuracy and precision test results for the six algorithms on UNSW-NB15. Each bar represents the result and standard deviation obtained from 100 repeated runs of the feature subset selected by the algorithm using the DT and RF classifiers. The experimental results show that the SABPIO algorithm improves the accuracy rate by 0.12% to 4.89% and the precision rate by 0.19% to 5.98% compared to

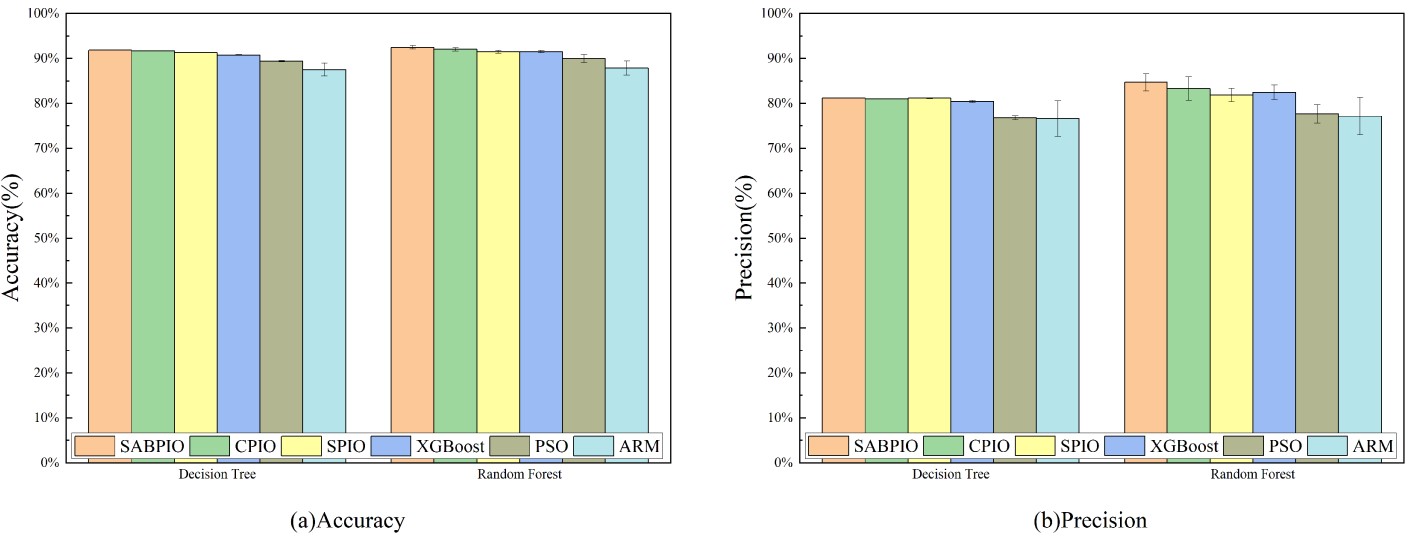

(a)Accuracy                (b)Precision

**Figure 7 (A and B) Accuracy and precision on the UNSW-NB15 dataset by DT and RF.**

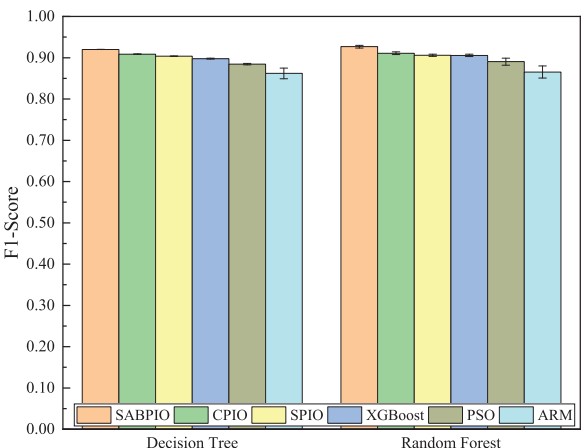

**Figure 8 F1-score on UNSW-NB15 dataset by DT and RF.**

other algorithms, for an equivalent number of iterations. It is important to consider both accuracy and precision rates, as well as other relevant indicators. The PSO and ARM algorithms are highly accurate but have low precision, indicating a higher likelihood of false predictions in samples identified as cyber-attacks. This tendency often leads to higher misclassification rates within the models, resulting in more instances of misclassifying normal traffic as attacks.

Figure 8 shows the mean and standard deviation of the F1-Score from 100 repeated experimental tests for the feature subsets selected by the SABPIO algorithm and other algorithms using the DT and RF classifiers. The results indicate that the F1-score achieved by SABPIO is 0.920 in the DT classifier and 0.927 in the RF classifier, demonstrating superior performance compared to the other five algorithms. Furthermore, the lower
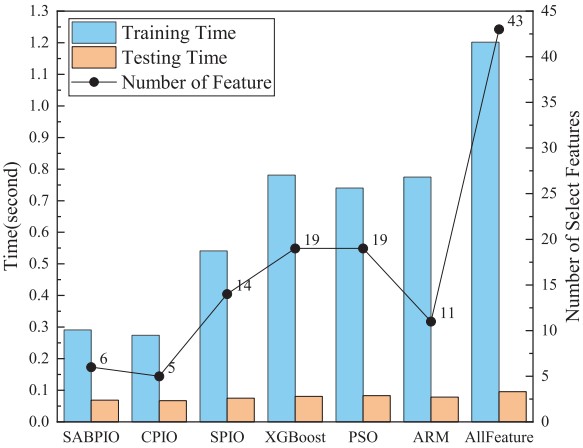

**Figure 9 Training and testing time and number of select features on the UNSW-NB15 dataset by RF.**

standard deviation highlights the improved performance of the SABPIO algorithm, indicating better consistency and stability. The selected feature subset demonstrates superior feature representation capabilities and heightened performance stability.

Figure 9 shows a comparison of training and testing times before and after feature selection for different feature subsets selected by various feature selection algorithms on UNSWNB15. The results demonstrate that the number and quality of features have a significant impact on the model's training and testing time. The SABPIO feature selection algorithm can significantly reduce model training time and improve efficiency without compromising detection results. The experiment evaluated the training time using the RF classifier. The training time of the RF classifier before feature selection was 1.21 s. After SABPIO feature selection, the training time reduced to 0.29 s, which is about 3.2 times faster than using all the features. Additionally, the testing time decreased from 0.096 to 0.068 s.

(2) Results of NSL-KDD

The NSL-KDD dataset was utilized to evaluate the detection performance of various algorithms including SABPIO, CPIO, SPIO, IG, PSO, and ARM. Figure 10 illustrates the DR and FAR of 100 repeated tests on DT and RF classifiers using the SABPIO algorithm and the feature subset selected by the recent feature algorithm. As shown in Fig. 10A, SABPIO outperforms the other DT classifiers with a DR of 90.2% (±1.3%), which is an improvement of approximately 3.6% compared to the next best CPIO. In Fig. 10B, the SABPIO algorithm prioritizes the balance of DR and FAR. Although the FAR is slightly higher compared to other algorithms such as SPIO, IG, PSO, and ARM, it still has a significant difference with the proposed algorithms in terms of DR. Similar to the DT classifier experiments, the RF classifier using the SABPIO algorithm showed significantly better DR means than the other algorithms in 100 repetitions, with slightly higher FAR. The proposed algorithm also maintained a low standard deviation, demonstrating the robustness and interpretability of the selected feature subset.

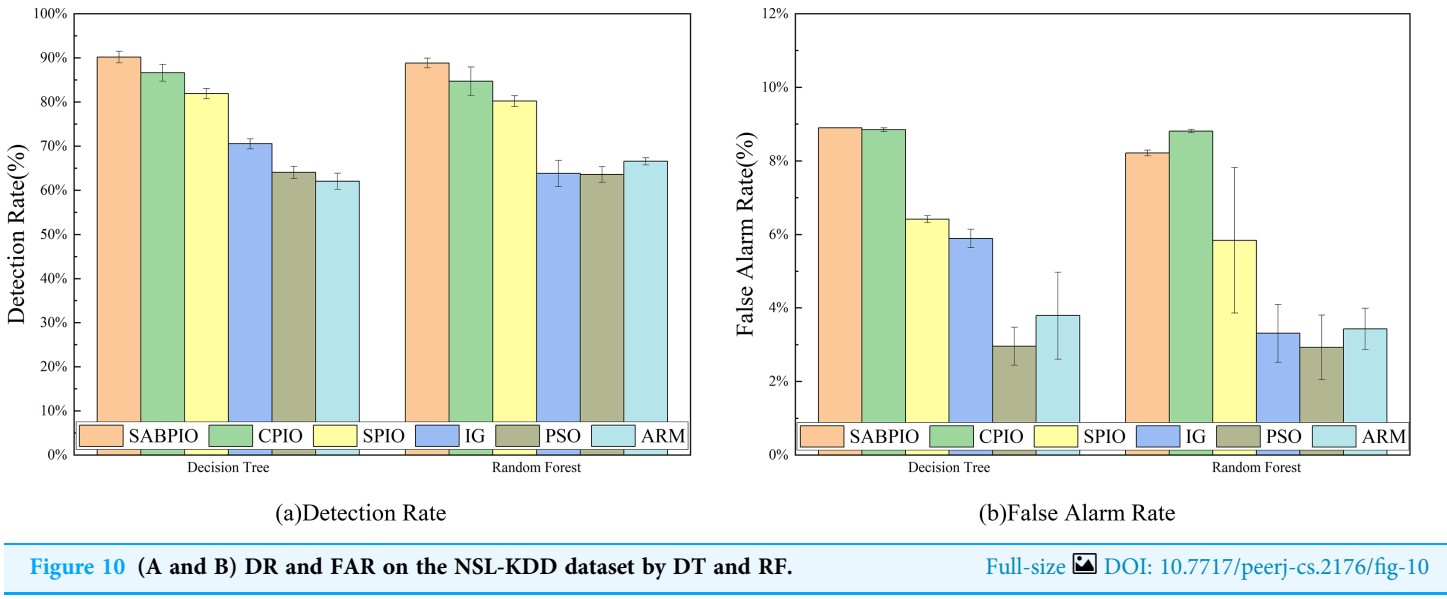

(a)Detection Rate

(b)False Alarm Rate

**Figure 10** **(A and B) DR and FAR on the NSL-KDD dataset by DT and RF.**

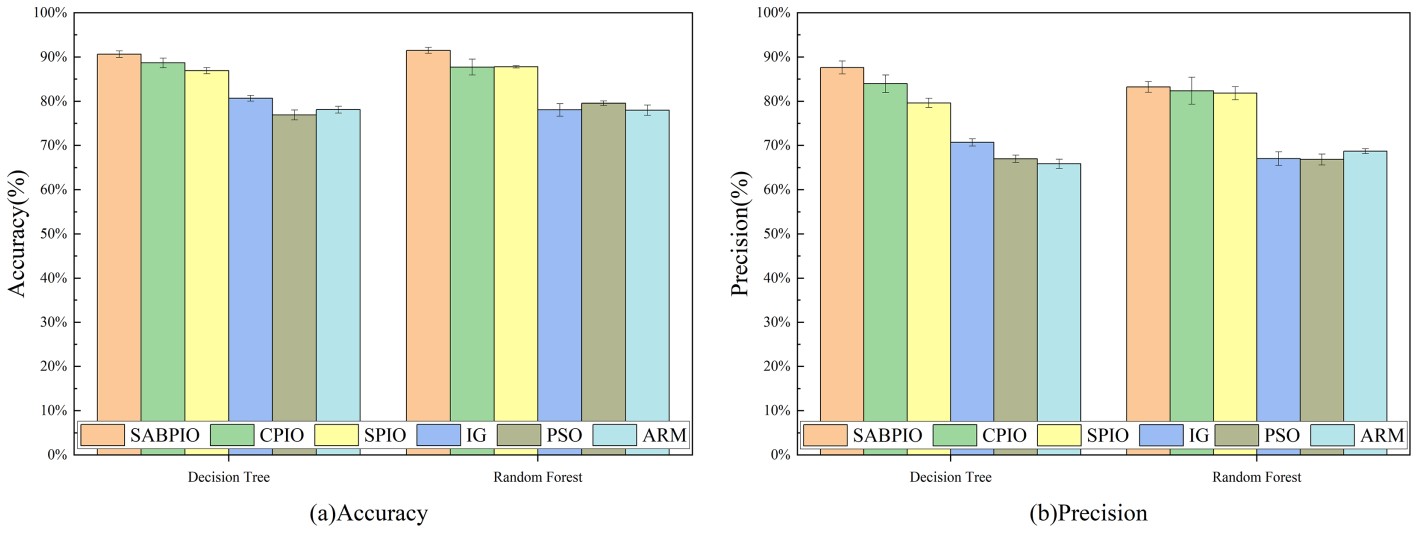

(a)Accuracy

(b)Precision

**Figure 11** **(A and B) Accuracy and precision on the NSL-KDD dataset by DT and RF.**

Figure 11 displays the results of 100 repeated experiments on NSL-KDD for the feature subsets selected by six feature selection algorithms. The experimental results indicate that the SABPIO algorithm outperforms the other five algorithms in terms of accuracy and precision, achieving 90.6% (±0.7%) and 91.5% (±0.6%) respectively, under the same number of iterations, whether using a DT classifier or an RF classifier. The accuracy of the other algorithms was 87.6% (±1.4%) and 83.2% (±1.2%).

Figure 12 displays the mean and standard deviation of the F1-Score from 100 repeated experimental tests on the DT and RF classifiers for the feature subsets selected by the

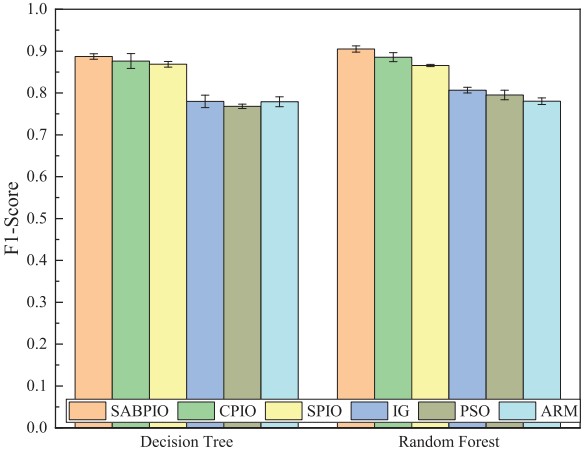

**Figure 12 F1-score on the NSL-KDD dataset by DT and RF.**

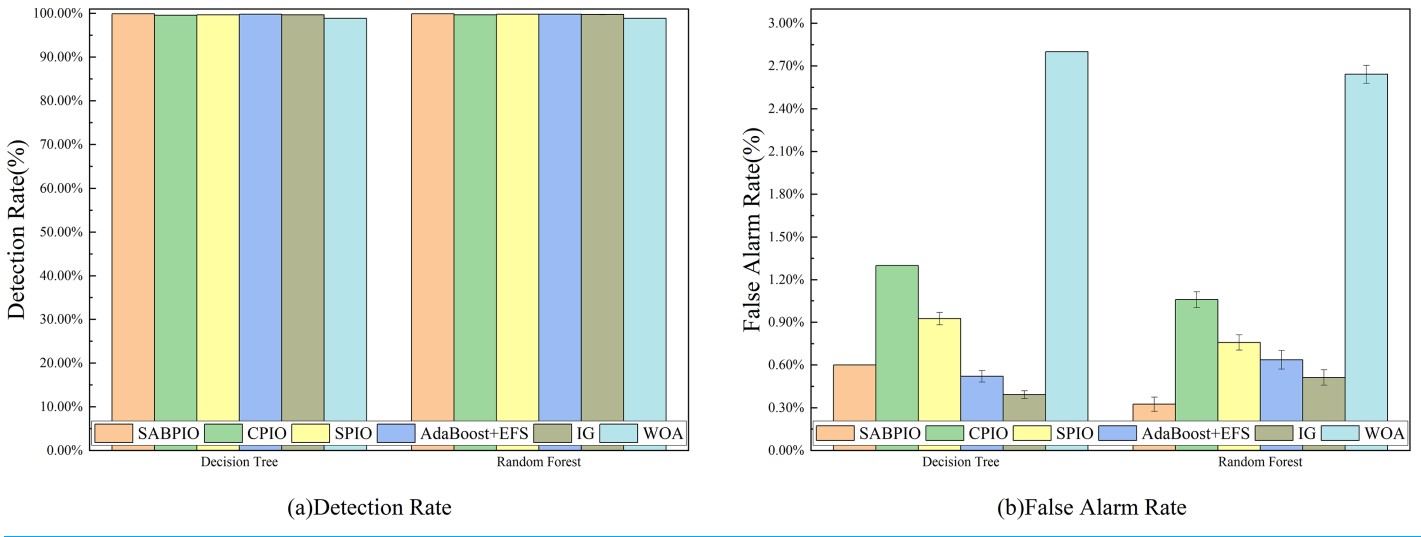

(a)Detection Rate

(b)False Alarm Rate

**Figure 13 (A and B) DR and FAR on the CIC-IDS-2017 dataset by DT and RF.**

SABPIO algorithm and other algorithms in the NSL-KDD dataset. The experimental results indicate that the F1-Score of SABPIO reaches 0.887 in the DT classifier and 0.904 in the RF classifier. SABPIO outperforms the other five algorithms with the best performance and a lower standard deviation.

(3) Results of CIC-IDS-2017

We selected a random sample of 20% of the data in the CIC-IDS-2017 dataset for experiments and tested the detection performance of SABPIO with CPIO, SPIO, AdaBoost+EFS, IG, and WOA algorithms. Figure 13 displays the DR and FAR of 100 repeated trial tests on DT and RF classifiers using the feature subset selected by the SABPIO algorithm and the remaining five feature algorithms. Figure 13A shows that SABPIO achieves higher DR than other feature selection algorithms, with 99.892% and

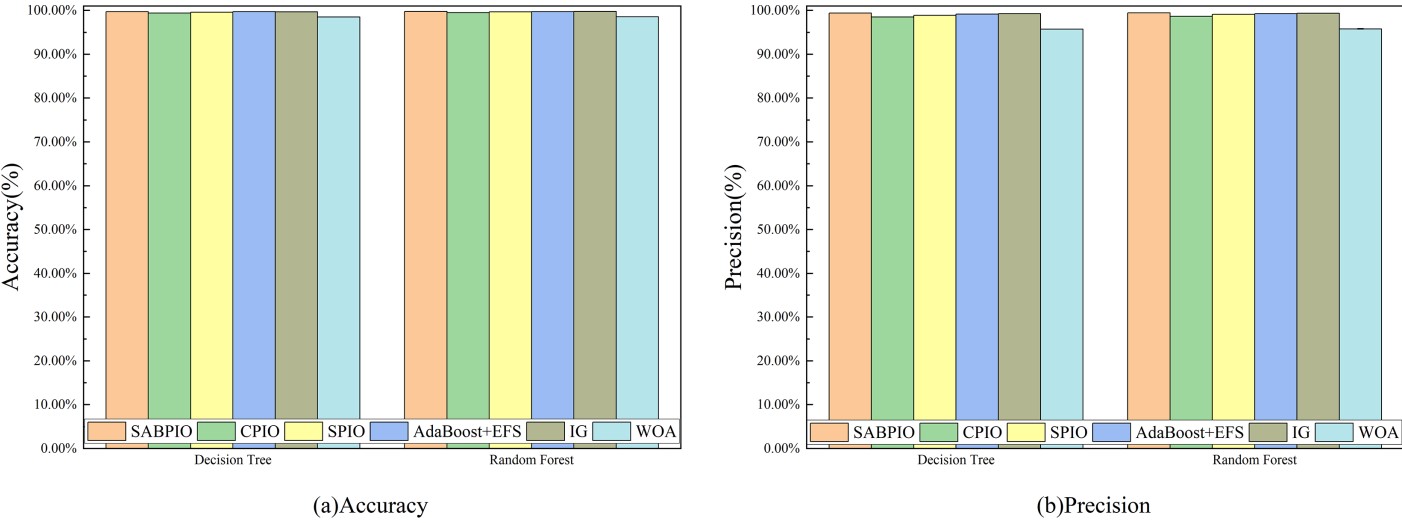

(a)Accuracy                (b)Precision

**Figure 14** (A and B) Accuracy and precision on the CIC-IDS-2017 dataset by DT and RF.  Full-size 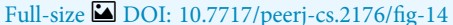 DOI: 10.7717/peerj-cs.2176/fig-14

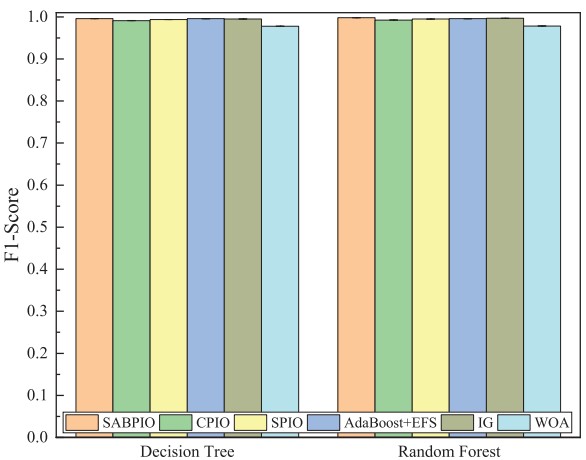

**Figure 15** F1-score on the CIC-IDS-2017 dataset by DT and RF.

99.897% in DT and RF classifiers, respectively. It is important to note that all evaluations are objective and based on empirical evidence. Figure 13B indicates that SABPIO is at the optimal level of FAR in both classifiers, except for a slightly higher FAR in the DT classifier compared to the IG algorithm.

Figure 14 displays the accuracy and precision test results of the feature subset selected by six feature selection algorithms on 20% of CIC-IDS-2017, repeated 100 times. The experimental results indicate that, under the same number of iterations, the SABPIO algorithm outperforms the other five algorithms in terms of accuracy and precision for both DT and RF classifiers, achieving 99.72%, 99.80%, and 99.38%, respectively, with an overall accuracy of 99.44%.

Figure 15 displays the mean and standard deviation of the F1-score from 100 repeated experimental tests on DT and RF classifiers for the feature subsets selected by the SABPIO

algorithm and other algorithms in 20% of the CIC-IDS-2017 dataset. The experimental results indicate that the F1-score of SABPIO reaches 0.996 in the DT classifier and 0.998 in the RF classifier, demonstrating superior performance and lower standard deviation compared to the other five algorithms.

## CONCLUSION

Network intrusion detection detects attacks by monitoring traffic. However, the large volume and high dimensionality of network data pose challenges to intrusion detection. Redundant and irrelevant features seriously affect detection performance. To address these, by incorporating mutation and simulated annealing into the map and compass operator, as well as introducing a population decay factor in the landmark operator phase. Experimental results indicate that the SABPIO algorithm effectively improves the detection rate and reduces false alarms, as well as training time.

However, it should be noted that SABPIO is subject to limitations that depend on the quality and completeness of the data. In the event that there are a significant number of missing or outlier values in the dataset, SABPIO may not be able to achieve optimal performance. In our future work, we will investigate how to improve the SABPIO algorithm to handle incomplete data. Meanwhile, the number of network attack samples and normal traffic samples is unbalanced in the actual network environment, so in further research, we will consider the impact of sample distribution imbalance on the feature selection algorithm.

### Funding

This work was supported by the China Higher Education Institution Industry-University-Research Innovation Fund (Nos. 2021FNB03001 and 2022IT020), the Stabilization Support Program of The Shenzhen Science and Technology Innovation Commission (No. 202311130110921001), and the Key Scientific Research Project of Higher Education Institutions of Henan Province (No. 24A520042). There was no additional external funding received for this study. The funders had no role in study design, data collection and analysis, decision to publish, or preparation of the manuscript.

### Grant Disclosures

The following grant information was disclosed by the authors:
China Higher Education Institution Industry-University-Research Innovation Fun: 2021FNB03001 and 2022IT020.
The Shenzhen Science and Technology Innovation Commission: 20231130110921001.
Higher Education Institutions of Henan Province: 24A520042.

### Competing Interests

Xiaohui Zhang is employed by Henan Xinda Wangyu Technology Co. Ltd.

## Author Contributions

- Wanwei Huang conceived and designed the experiments, performed the experiments, analyzed the data, authored or reviewed drafts of the article, and approved the final draft.
- Haobin Tian conceived and designed the experiments, performed the experiments, analyzed the data, performed the computation work, prepared figures and/or tables, authored or reviewed drafts of the article, and approved the final draft.
- Sunan Wang conceived and designed the experiments, analyzed the data, authored or reviewed drafts of the article, and approved the final draft.
- Chaoqin Zhang analyzed the data, prepared figures and/or tables, and approved the final draft.
- Xiaohui Zhang performed the experiments, performed the computation work, prepared figures and/or tables, and approved the final draft.

## Data Availability

The source code is available in the Supplemental File.

The intrusion detection datasets UNSW-NB15, NSL-KDD and CICIDS2017 that were used in this study are available at, respectively:

- UNSW-NB15: https://research.unsw.edu.au/projects/unsw-nb15-dataset. Compiled by the Cyber Range Lab of the Australian Centre for Cyber Security, this dataset is intended for network intrusion detection, featuring modern attack types in a realistic network traffic scenario.

- NSL-KDD: https://web.archive.org/web/20150205070216/http://nsl.cs.unb.ca/NSL-KDD/ (The original link was: https://www.unb.ca/cic/datasets/nsl.html). An improved version of the KDD'99 dataset for network-based intrusion detection systems. It removes redundant records, thus providing a more effective dataset for training and evaluating IDS models.

- CICIDS2017: https://www.unb.ca/cic/datasets/ids-2017.html. Developed by the Canadian Institute for Cybersecurity (CIC), provides a modern platform for IDS/IPS system evaluation. It includes a diverse set of traffic scenarios and a variety of attacks, making it an effective tool for IDS model training and evaluation.

## Supplemental Information

Supplemental information for this article can be found online at http://dx.doi.org/10.7717/peerj-cs.2176#supplemental-information.

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
