# Peer review of "Integration of simulated annealing into pigeon inspired optimizer algorithm for feature selection in network intrusion detection systems"

_PeerJ Computer Science, doi:10.7717/peerj-cs.2176_

## Round 0.1 · original submission · Major Revisions

Dear authors,

Thank you for submitting your article. Feedback from the reviewers is now available. It is not recommended that your article be published in its current format. However, we strongly recommend that you address the issues raised by the reviewers, especially those related to readability, experimental design and validity, and resubmit your paper after making the necessary changes. When submitting the revised version of your article, it will be better to also address the following:

1. The research gaps and contributions with respect to the problem and the methodology should be clearly summarized in the Introduction section. Metaheuristic based methods already are used in feature engineering. Please evaluate how your study is different from others in the related work section.
2. Organization of the paper may be given at the end of Introduction section.
3. Keyword should be written in alphabetical order.
4. The analysis and configurations of experiments should be presented in detail for reproducibility. It is convenient for other researchers to redo your experiments and this makes your work easy acceptance. A table with parameter settings for experimental results and analysis should be included in order to clearly describe them. The values of the parameters of the algorithms selected for comparison should be mentioned.
5. The authors should clarify the pros and cons of the methods. What are the limitation(s) methodology(ies) adopted in this work? Please indicate practical advantages, and discuss research limitations.
6. Appropriate references should be used for the equations. They seem they are firstly proposed in this paper.
7. Blank character should be correctly used in the text.

Best wishes,

Reviewer 1 ·

Basic reporting

-The writing language of the article should be reviewed and some errors should be corrected.
- The related studies section is insufficient. Studies on feature selection are insufficient. Studies on intrusion detection are also presented in only one table. Not enough information is given about these studies.
-The resolution of the location representation of pigeons (Figure 2) is quite low.

Experimental design

- Why is the weight factor set to 0.0075 in the fitness function? Why was the pigeon number 128 chosen for the working parameters of the algorithm?
-The effectiveness of the SAPIO algorithm was compared with the SPIO algorithm on different data sets. If SPIO is a version of PIO, what it is should be stated in the article.
-What is meant by the concept of rational computation in the expression "The paper presents a method that selects the most representative feature subset through rational computation." Can the real computation situation also be evaluated?

Validity of the findings

- What is meant by the sentence "Hence, the significance of feature selection algorithms in NIDS cannot be overstated"? Then why were feature selection methods applied for the NIDS problem in this study? This sentence should be changed or removed.
- Information about the motivation of the study was not found in the paper.
-There are ambiguous statements in the key contributions of the article. It is not stated how the shortcomings of fast convergence and susceptibility to local optima were eliminated.
-How the binary version of the proposed PIO algorithm is proposed. Was a transfer function used? (v-shaped, s-shaped, etc)
-Why was the Pigeon Inspired Optimizer Algorithm preferred for feature selection in the Network Intrusion Detection Systems problem? Why not PSO or GA?

Additional comments

In this study, a method is proposed to evaluate the performance of the system by making changes to the Pigeon Inspired Optimizer algorithm for Network Intrusion Detection. We thank the authors for their work. However, there are shortcomings in the article taking into account the evaluations I have made under different headings.

Reviewer 2 ·

Basic reporting

This paper proposes a feature selection algorithm for network intrusion detection systems that uses an improved binary pigeon-inspired optimizer (SABPIO) algorithm to tackle the challenges posed by the high dimensionality and complexity of network traffic.

The paper is well-written, clear, and easy to follow. The figures provided in the paper are of a high quality, aiding the reader's understanding. However, it may be helpful to note that Figure 2 requires improvement as the quality of presentation is not the same as the other figures.

Table 1 indicates that when using the XGBoost method, the total number of features for UNSW-NB15 is 43, whereas in wrapper methods, it is stated that the total number of features is 42. Is this difference a typo or there is a reason behind it. Furthermore, the total number of features given in Table 1 does not match the ones in given Tables 2,3, and 4.

The literature review in the paper categorizes existing studies based on feature selection methods, specifically wrapper, filter, and hybrid approaches. While this classification is informative, the paper would benefit from placing a stronger emphasis on the wrapper method and cite more recent research in this area to enhance the related work section.

A minor comment on the abstract as it contains a redundant statement regarding the reduction of system training time. The authors mentioned that the process notably reduces system training time and also decreases both training and testing times by at least 3.2 and 0.3 times, respectively. To avoid repetition, it is recommended to rephrase the sentence.

Experimental design

The problem statement is presented with clarity, providing a comprehensive overview of the issues being addressed. Moreover, the contributions of the study are outlined concisely and explicitly, offering a clear understanding of the significance of the research.

The methodology section of the paper consists of several subsections. While it is well-presented, it lacks clarity in distinguishing between the author's contributions and existing literature. It would be beneficial for the authors to explicitly indicate which parts and equations are their own contributions and provide a rationale behind these equations. For instance, including the rationale behind the fitness function formula if introduced by the authors would enhance the understanding of the methodology.

Validity of the findings

The obtained results demonstrate an improvement over the existing literature. However, the experiment section lacks clarity in explaining the method of splitting the data into training and testing datasets, as well as the specific percentage for each.

---

## Round 0.2 · accepted · Accept

Dear authors,

Thank you for the revision and for clearly addressing all the reviewers' comments. I confirm that the paper is improved. Your paper is now acceptable for publication in light of this revision.

Best wishes,


Reviewer 1 ·

Basic reporting

No comment

Experimental design

No comment

Validity of the findings

No comment

Additional comments

The authors made the necessary revisions to the article according to my suggestions.

Reviewer 2 ·

Basic reporting

no comment

Experimental design

no comment

Validity of the findings

no comment